# An Improved Convolutional Layer Based on Stochastic Masked Kernel for Ship Target Detection

Jiachang Zhang
Navigation College
Dalian Maritime University
Dalian, China
1120231093@dlmu.edu.cn

Yi Zuo
Navigation College
Dalian Maritime University
Dalian, China
zuo@dlmu.edu.cn

Junhao Jiang
Navigation College
Dalian Maritime University
Dalian, China
jiangjunhao@dlmu.edu.cn

Licheng Zhao
Navigation College
Dalian Maritime University
Dalian, China
zhaolichengzx@dlmu.edu.cn

*Abstract*—**Ship target detection is crucial for ensuring maritime safety. To tackle the issues of limited recognition accuracy and inadequate generalization in existing detection models, this thesis presents an enhanced model: YOLOv5s-Mask, which incorporates an Improved Convolutional Layer based on Stochastic Masked Kernels. This novel approach aims to boost both recognition accuracy and the model's ability to generalize across various scenes. We introduce the concept of the Stochastic Masked Kernel and develop the Masked SPPF (Masked Spatial Pyramid Pooling) module to improve the model's detection performance and robustness, particularly for small and densely packed targets. Additionally, we incorporate the SE (Squeeze-and-Excitation) attention mechanism to further refine recognition accuracy while keeping the model lightweight. Experimental results show that YOLOv5s-Mask achieves a 1.26% improvement in mAP and a 5.71% increase in mAP0.5-0.95 compared to the original YOLOv5s. This demonstrates the model's significant potential for real-world ship target detection applications.**

*Keywords—Target Detection, YOLOv5s, Stochastic Masked Kernel, Ship Target Recognition*

## I. INTRODUCTION

The growing global trade has increasingly highlighted the importance of the shipping industry[1]. With the rising number of ships and the evolving shipping scenarios, ship target detection has become crucial for maritime traffic management and shipping safety. In complex shipping environments—such as ports with high traffic volumes and narrow waterways—traditional target detection algorithms often struggle with limited generalization and accuracy. Therefore, developing a ship target detection method that excels in accuracy, generalization, and robustness is of significant importance.

Target detection algorithms are typically divided into two main categories: two-stage algorithms and one-stage algorithms. However, traditional two-stage algorithms, such as Faster R-CNN[2] and the R-CNN[3] series, face challenges including high computational resource demands, diverse ship sizes and shapes, and limited feature extraction generalization. To address these issues, one-stage algorithms like YOLO[4] and SSD[5] have gained prominence and application in ship target detection. One-stage algorithms offer several advantages: they are typically more efficient in inference as they directly predict target locations and classes without complex candidate region extraction; they can be optimized through end-to-end training, facilitating adaptation to ships of various sizes and shapes; and some one-stage algorithms, such as YOLOv5, enhance detection in complex backgrounds by employing techniques like SPPF to integrate multi-scale feature information.

Many scholars have proposed numerous algorithms before and after. Liu et al.[6] proposed an augmented CNN to improve the capability for detecting ships in different weather conditions. Li et al.[7] suggested an enhanced Faster R-CNN model, which delivers strong performance on the SSDD dataset. Lei et al.[8] introduced the SRSDD dataset for high-resolution SAR rotating ship detection, which includes information on both ship categories and ship angles. Qi et al.[9]employs a scene narrowing technique to integrate the target area localization network with the Faster R-CNN convolutional neural network into a multi-layered narrowing network. This approach reduces the target detection search scale and enhances the computational speed of Faster R-CNN. Dong et al.[10] proved that the YOLOv5 algorithm is effective in target detection. Yue et al.[11] designed a feature extraction network suitable for SAR ship target detection by combining VGGNet with dilated convolution. Zheng[12] replaced the YOLOv4 backbone network with MobileNetV1 in distance measurement to enhance detection speed during the recognition phase. Zhang[13] introduced CA-Ghost and C3Ghost for feature extraction in the backbone and neck layers, respectively, to enhance detection performance while optimizing the model. Zhang et al.[14] replaced DarkNet53 with DarkNet19 in YOLOv3, resulting in improved speed.T. Li et al.[15] proposes a BLIoU method based on the Broad Learning System , which significantly enhances target tracking performance through Intersection over Union network-based scale and drift correction, while featuring short training time and strong portability.T. Li et al.[16] proposed a method that integrates Siamese networks with the Broad Learning System, enhancing the accuracy and adaptability of target tracking by

rapidly learning target features online. However, relatively few studies have been conducted on complex environments and overlapping targets in ship target detection tasks, and there are cases of missed or false detection. Therefore, enhancing the robustness and generalization of target detection models in ship target detection remains a challenging issue that warrants further research.

Our thesis introduces a Stochastic Masked Kernel model based on YOLOv5s, referred to as YOLOv5s-Mask. The proposed model involves modifying the convolution layer to incorporate a Stochastic Masked Kernel, built upon SPPF, to enhance robustness and generalization capability while reducing false detections. Additionally, the SE attention mechanism is incorporated before the SPPF to enhance the accuracy. Comparative tests were conducted on four models: the original YOLOv5s, YOLOv5s-Atrous, YOLOv5s-SE, and YOLOv5s-Mask (our model). The results indicate that our model achieved the best performance, with improvements of 1.09% in precision, 2.46% in recall, 1.26% in mAP0.5, and 5.71% in mAP0.5-0.95. The new model effectively enhances precision and recall, addresses issues related to generalization and robustness, and performs well in complex shipping environments.

Section 2 details the methodology for the Masked SPPF and the incorporation of the SE attention mechanism. Section 3 highlights the benefits of the proposed model in terms of precision, recall, and generalization through comparative experiments. Section 4 outlines potential research directions, including improved mask generation design and development of more practical models.

## II. METHODOLOGY

### A. YOLOv5s algorithm

YOLOv5s[17] is part of the YOLOv5 series. The strength of the YOLOv5s structure is its integration of a compact model size with efficient feature extraction capabilities, which is suitable for real-time object detection application scenarios that require fast processing speed and high accuracy, so this paper is based on the YOLOv5s model for optimization.

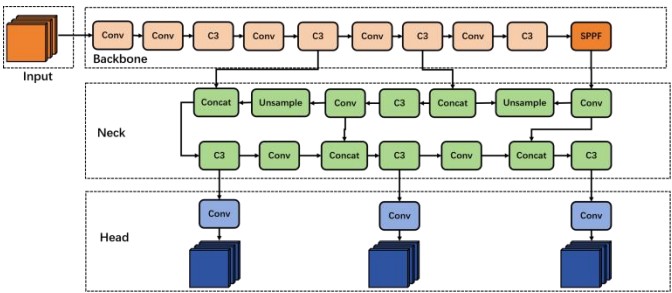

Fig. 1 Network structure of YOLOv5s

### B. Masked SPPF

Spatial Pyramid Pooling (SPP) was first introduced by He et al.[18] in 2014 to address the issue of fixed-size input requirements imposed by fully connected layers when dealing with variable-size inputs in convolutional neural networks. The core concept of SPP is to apply pooling operations at multiple scales on a fixed-size feature map, enabling multi-scale representation of objects within the input image or feature map. SPP achieves this by partitioning the feature map into grids of different sizes and applying pooling kernels of different dimensions to each grid. This approach captures feature information at different scales without necessitating changes to the network structure. Additionally, the feature vectors produced by SPP have a fixed size, making them suitable for subsequent fully connected layers or classifiers, regardless of the variations in input size.

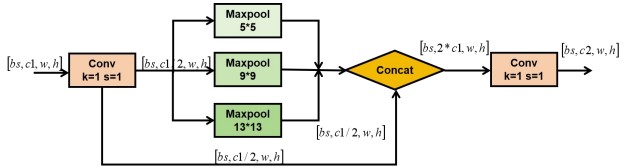

Fig. 2 Network structure of SPP

Spatial Pyramid Pooling with Feature Fusion (SPPF) is an advanced version of SPP. Unlike its predecessor, SPPF introduces a feature fusion mechanism that enhances the network's capacity to represent multi-scale objects by integrating feature information across various scales and pooling layers. This fusion process helps to mitigate information loss and improves the model's stability and generalization in complex scenes. As a result, SPPF achieves higher precision and accuracy in object detection and image recognition tasks. It is particularly effective for handling images with significant scale variations or complex backgrounds.

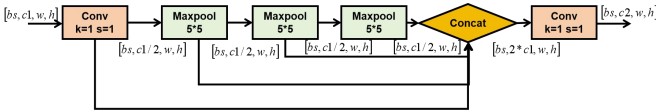

Fig. 3 Network structure of SPPF

The benefits of incorporating random masked convolutions into SPPF are as follows:

*1) Feature Enhancement: Masked convolutions increase feature diversity, while SPPF extracts multi-scale features. The combination of the two can provide a more comprehensive representation of the target, thereby improving detection accuracy.*

*2) Generalization Capability Improvement: The randomness of masked convolutions helps reduce overfitting, and SPPF enhances the model's adaptability to different target sizes and proportions. Together, they improve the model's generalization performance across various environments and conditions.*

*3) Efficiency and Complexity Balance: By rationally designing the network structure and combining the two, it is possible to ensure model performance while minimizing*

*computational complexity and improving computational efficiency.*

The Stochastic Masked Kernel is implemented by setting the mask_prob parameter to 0.1. This involves creating a stochastic binary mask matrix that matches the dimensions of the convolution kernel. This binary mask is then applied to the convolution kernel within the SPPF. The feature maps are subsequently convolved with the modified kernel. Specifically, Stochastic Masked Kernel is introduced into the convolution process after the feature map has been processed by the SPPF maximum pooling layer. By randomly altering portions of the convolution kernel, the model's sensitivity and robustness to small and dense targets are improved.

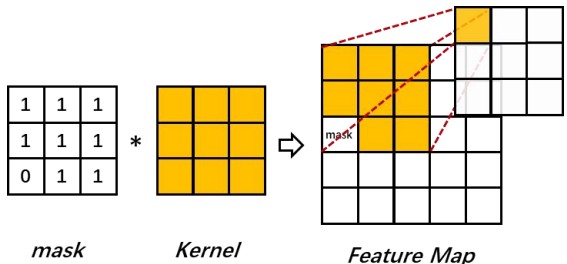

Fig. 4 Schematic of Stochastic Masked Kernel

$$O_t^{k=s} \xrightarrow{F} Random(k) \cdot K \qquad (1)$$

Adding the Stochastic Masked Kernel to the SPPF architecture offers several advantages. By randomly generating masks, the shape and position of the convolution kernel can be changed. This variability enhances the model's sensitivity to dense targets, thereby enhancing accuracy. The introduction of randomness increases the model's versatility, enabling it to behave differently each time it is trained or evaluated. This is particularly beneficial for handling complex scenarios, noisy interferences, or datasets with variations, thereby boosting the model's robustness. Additionally, the Stochastic Masked Kernel acts as a regularization technique, reducing the risk of overfitting. By introducing randomness during training, the model's generalization ability is enhanced. While SPPF already incorporates multiscale feature fusion, the Stochastic Masked Kernel further refines multiscale feature representation. The use of masks introduces variations in feature representations across different locations and scales, which helps capture a broader range of semantic information.

*C. SE Attention Mechanism*

The SE (Squeeze-and-Excitation) Attention Mechanism[19] is a channel attention module that consists of two primary phases: Squeeze and Excitation. It is commonly used in visual modeling to enhance feature representations. The SE module operates in a plug-and-play manner, allowing it to be seamlessly integrated into existing architectures. It enhances the input feature maps at the channel level while preserving the original size of the input feature maps in the final output.

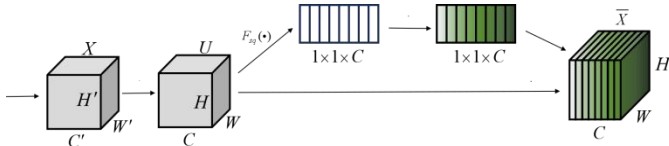

Fig. 5 SE Block structure

SE Block structure is depicted in Fig. 2, firstly, $X \in R^{H' \times W' \times C'}$ is mapped to the feature map $U \in R^{H \times W \times C}$ ,by the convolution operation $F_{tr}$ In this process, the set of convolution kernels is denoted by $V = [v_1, v_2, ..., v_C]$ , and the output can be represented as $U = [u_1, u_2, ..., u_C]$ , then:

$$u_c = v_c * X = \sum_{s=1}^{C'} v_c^s * x^s \qquad (2)$$

The * is convolution operation.

$u_c \in R^{H \times W}$ , $v_c = [v_c^1, v_c^2, ...v_c^{C'}]$ , $X = [x^1, x^2, ..., x^{C'}]$ , $v_c^s$ is a two-dimensional convolution kernel, meaning that one channel of $v_c$ interacts with the matching channel of $X$.

Squeeze phase: to consider the information from each channel in the output feature map, it compresses the global spatial elements into a channel descriptor $Z_c$ by means of global average pooling. It is usually done to Minimize computation and parameters. The formula for this is:

$$z_c = F_{sq}(u_c) = \frac{1}{H \times W} \sum_{i=1}^{H} \sum_{j=1}^{W} u_c(i,j) \qquad (3)$$

Excitation Phase: It uses a gating mechanism with Sigmoid activation to fully capture channel dependencies. This mechanism learns the importance weights of each channel using a small feed-forward neural network, typically a fully connected network. The formula for this is:

$$s = F_{ex}(z, W) = \sigma(g(z, W)) = \sigma(W_2 \delta(W_1 z)) \qquad (4)$$

where $\delta$ is the ReLU function, $W_1 \in R^{\frac{C}{r} \times C}$ , $W_2 \in R^{\frac{C}{r} \times C}$ .

Scale phase: finally, the learned weights are multiplied with the feature map and used to weight the channels with higher importance. This process allows the network to adaptively Concentrate on key features during the learning process, thus improving the overall performance. The SE Block's final output is achieved by rescaling $U$ with $s$:

$$\overline{x_c} = F_{scale}(u_c, s_c) = s_c u_c \qquad (5)$$

In this model, the SE module is inserted into the convolutional layer preceding the SPPF layer. This integration allows the importance weights of each feature map channel to be learned by the model, enabling the network to focus more on features that are crucial for object classification and bounding box prediction. As a result, the information in the feature maps is utilized more efficiently, improving precision and stability in target detection. Notably, in the compact

YOLOv5s model, the SE module significantly boosts performance with minimal additional computational cost.

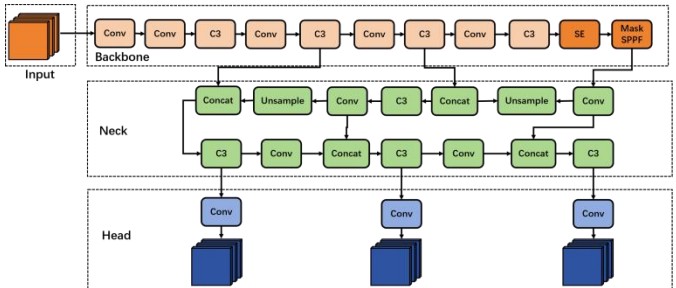

Fig. 6 Improved YOLOv5s network structure

## III. EXPERIMENT AND RESULTS

### A. Data sets

The Seaships dataset is used, an open dataset specifically designed for ship detection and recognition. It includes a substantial number of ship targets captured in aerial or satellite imagery, with ships appearing in various sizes, orientations, and lighting conditions over water bodies such as oceans and harbors. The dataset comprises 7,000 images, with 80% allocated for training the model and 20% reserved for validation to assess detection performance.

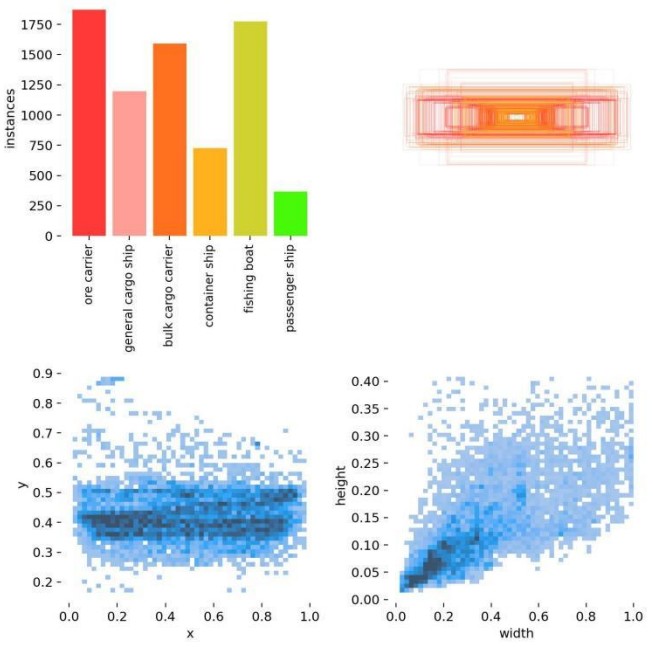

Fig. 7 Data distribution for the seaships dataset

### B. Evaluation metrics and experiments

The mAP is selected as the evaluation metrics AP and mAP are calculated by formula 6 and formula 7.

$$AP = \int_0^1 P\,dR \tag{6}$$

$$mAP = \frac{\sum_{i=1}^{N} AP_i}{N} \tag{7}$$

N represents the total count of categories, precision rate P and recall rate R are calculated from formula 8 and formula 9.

$$P = \frac{TP}{TP + FP} \tag{8}$$

$$R = \frac{TP}{TP + FN} \tag{9}$$

TABLE I. RESULTS OF ABLATION EXPERIMENTS

| Mould | P | R | mAP0.5 | mAP0.5-0.95 |
|---|---|---|---|---|
| YOLOv5s | 98.03% | 96.03% | 97.73% | 75.88% |
| YOLOv5s-Atrous | 97.56% | 96.40% | 98.69% | 76.49% |
| YOLOv5s-SE | 98.11% | 96.78% | 98.70% | 76.52% |
| YOLOv5s-Mask | 99.12% | 98.49% | 98.99% | 81.59% |

As shown in TABLE I., the proposed model enhances precision by 1.09%, recall by 2.46%, mAP0.5 by 1.26%, and mAP0.5-0.95 by 5.71% in contrast to the conventional YOLOv5s. The mAP0.5-0.95 metric is widely used in target detection as it comprehensively assesses a model's accuracy and generalization across various Intersection over Union (IoU) thresholds. This makes it a valuable tool for evaluating and comparing algorithm performance. Although the new model shows only marginal improvements in precision and recall, it demonstrates superior generalization and robustness.

## IV. CONCLUSION AND FOLLOW-UP

To address the limitations of insufficient generalization and robustness in ship target detection models under complex conditions, this paper proposes an enhanced YOLOv5s model incorporating Stochastic Masked Kernel and SE attention mechanism. This improvement involves replacing the standard convolution in the original SPPF with Stochastic Masked Kernel and inserting the SE attention mechanism before this module. Results show that our model achieves a 1.09% increase in precision, a 2.46% increase in recall, a 1.26% increase in mAP0.5, and a 5.71% increase in mAP0.5-0.95 on the Seaships dataset. Overall, the proposed model exhibits excellent performance, delivering high precision along with improved generalization and robustness in complex shipping environments.The model proposed in this paper provides robust technical support for enhancing the efficiency of marine traffic safety supervision in busy and ever-changing port and waterway environments.

Future work will focus on enhancing model predictability, accuracy, and efficiency, including further optimization of the stochastic mask generator and exploration of more effective stochastic mask generation techniques. Additionally, expanding and refining the ship target detection dataset will be pursued to boost the model's ability to generalize and its effectiveness in practical applications.

ACKNOWLEDGMENT

This work was supported in part by the National Natural Science Foundation of China (grant nos. 52131101 and 51939001), the Liao Ning Revitalization Talents Program (grant no. XLYC1807046), and the Science and Technology Fund for Distinguished Young Scholars of Dalian (grant no. 2021RJ08).

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
