# OpenReview forum: "An Improved Convolutional Layer Based on Stochastic Masked Kernel for Ship Target Detection"
_IEEE.org/ICIST/2024/Conference — IEEE ICIST 2024 Conference Submission_

### Official Review · Reviewer_7C2g · 2024-08-22
**This paper introduce the concept of the Stochastic Masked Kernel and develop the Masked Spatial Pyramid Pooling module to improve the model's detection performance and robustness, particularly for small and densely packed targets. Incorporate the Squeeze-and-Excitation attention mechanism to further refine recognition accuracy while keeping the model lightweight.**

**Rating:** 8
**Confidence:** 4

**Review:**

This paper introduce the concept of the Stochastic Masked Kernel and develop the Masked Spatial Pyramid Pooling module to improve the model's detection performance and robustness, particularly for small and densely packed targets. Incorporate the Squeeze-and-Excitation attention mechanism to further refine recognition accuracy while keeping the model lightweight. In general, this work is well organized and appears potentially interesting, it can be accepted with a little modification.
1.	To ensure the visual quality is optimal, it is recommended that the author provide higher resolution images to clearly showcase the relevant details and information.
2.	What are the future research directions outlined in this article?
3.	What are the innovative aspects of this system compared to others?
4.	Why choose Spatial Pyramid Pooling (SPP) in this paper?

---

### Official Review · Reviewer_KaTB · 2024-08-24
**Review Comments for Manuscript No. 38**

**Rating:** 7
**Confidence:** 4

**Review:**

1. The consistency of formula formatting, including symbols and fonts, needs to be thoroughly checked and revised.

2. The abbreviation "YOLOv5s-MaskSPPF-SE" is somewhat lengthy and could hinder readability. I suggest using a more concise abbreviation or a simplified alias to improve clarity and ease of reference throughout the manuscript.

3. It would be better to list the contributions in bullet points to highlight the key aspects more effectively.

4. The text size within the images is inconsistent, and some text becomes unclear when enlarged. This should be corrected to ensure readability.

5. The reference formatting is not uniform and should be standardized.

6. The abstract and content mention "Stochastic Masked Kernel" and "Masked SPPF" as core innovations of the model. What challenges do these innovations address? The manuscript should provide a more detailed explanation of the challenges these innovations overcome.

---

### Official Review · Reviewer_Wovn · 2024-08-25
**An Improved Convolutional Layer Based on Stochastic Masked Kernel for Ship Target Detection**

**Rating:** 9
**Confidence:** 3

**Review:**

This paper addresses a significant research problem with rich and engaging content. It proposes an enhanced model to tackle the issues of limited detection accuracy and insufficient generalization in existing ship target detection models. The experimental results validate the effectiveness of the proposed algorithm.

---

### Decision · Program_Chairs · 2024-09-06

Accept (Oral)